# A Novel HDL-Mimetic Peptide HM-10/10 Protects RPE and Photoreceptors in Murine Models of Retinal Degeneration

**DOI:** 10.3390/ijms20194807

**Published:** 2019-09-27

**Authors:** Feng Su, Christine Spee, Eduardo Araujo, Eric Barron, Mo Wang, Caleb Ghione, David R. Hinton, Steven Nusinowitz, Ram Kannan, Srinivasa T. Reddy, Robin Farias-Eisner

**Affiliations:** 1Department of Obstetrics and Gynecology, David Geffen School of Medicine, University of California at Los Angeles, Los Angeles, CA 90095, USA; fsu@mednet.ucla.edu; 2Department of Pathology, Keck School of Medicine, University of Southern California, Los Angeles, CA 90033, USA; ChristineK.Spee@med.usc.edu (C.S.); dhinton@usc.edu (D.R.H.); 3Jules Stein Eye Institute, University of California at Los Angeles, Los Angeles, CA 90095, USA; araujo@jsei.ucla.edu (E.A.); CGhion@hotmail.com (C.G.); nusinowitz@jsei.ucla.edu (S.N.); rkannan@doheny.org (R.K.); 4The Stephen J. Ryan Initiative for Macular Research, Doheny Eye Institute, Los Angeles, CA 90033, USA; ebarron2@doheny.org (E.B.); mwang@doheny.org (M.W.); 5Department of Ophthalmology, University of Southern California, Los Angeles, CA 90033, USA; 6Department of Medicine, David Geffen School of Medicine, University of California at Los Angeles, Los Angeles, CA 90095, USA; 7Department of Molecular and Medical Pharmacology, David Geffen School of Medicine, University of California at Los Angeles, Los Angeles, CA 90095, USA; 8Department of Obstetrics and Gynecology, School of Medicine, Creighton University, Omaha, NE 68178, USA

**Keywords:** retinal pigment epithelium (RPE), sodium iodate, geographic atrophy, retinitis pigmentosa, apoptosis, HDL-mimetic peptide, OCT, retinal function

## Abstract

Age-related macular degeneration (AMD) is a leading cause of blindness in the developed world. The retinal pigment epithelium (RPE) is a critical site of pathology in AMD. Oxidative stress plays a key role in the development of AMD. We generated a chimeric high-density lipoprotein (HDL), mimetic peptide named HM-10/10, with anti-oxidant properties and investigated its potential for the treatment of retinal disease using cell culture and animal models of RPE and photoreceptor (PR) degeneration. Treatment with HM-10/10 peptide prevented human fetal RPE cell death caused by *tert*-Butyl hydroperoxide (*t*BH)-induced oxidative stress and sodium iodate (NaIO_3_), which causes RPE atrophy and is a model of geographic atrophy in mice. We also show that HM-10/10 peptide ameliorated photoreceptor cell death and significantly improved retinal function in a mouse model of *N*-methyl-*N*-nitrosourea (MNU)-induced PR degeneration. Our results demonstrate that HM-10/10 protects RPE and retina from oxidant injury and can serve as a potential therapeutic agent for the treatment of retinal degeneration.

## 1. Introduction

Age-related macular degeneration (AMD) is characterized by progressive degeneration of the macular region of the retina resulting in loss of central vision. It causes damage to the central part of the retina, resulting in the loss of the light-sensitive tissue at the back of the eye. AMD is the leading cause of irreversible blindness in people aged 50 or older in the developed world and there is no complete cure for AMD [1]. The key to reducing the AMD-related vision loss is to be able to initiate therapies that could halt or slow down the progression of AMD.

High density lipoprotein (HDL) is an important mediator of lipid homeostasis. HDL and HDL associated molecules provide a number of protective functions including anti-inflammatory, antioxidant, anti-microbial, and innate immunity in multiple cell types and animal models [2]. HDL mimetic peptides have shown efficacy in a number of animal models of disease and demonstrate properties that make them attractive as potential therapeutic agents [3,4,5,6,7]. We developed a novel chimeric high density lipoprotein; mimetic peptide named HM-10/10. The arginine-rich cationic domain of human apoE {([141–150] hApoE) L-R-K-L-R-K-R-L-L-R}, was added to an apoJ mimetic, named G * peptide {L-V-G-R-Q-L-E-E-F-L, corresponding to amino acids 113 to 122 in apoJ ([113,122] apoJ)}, to form HM-10/10 peptide containing 20 amino acids with the sequence of L-R-K-L-R-K-R-L-L-R-L-V-G-R-Q-L-E-E-F-L.

Our aim in this study was to evaluate whether HM-10/10 peptide is beneficial in protecting retinal pigment epithelium (RPE) and photoreceptors from oxidant induced cell death under in vitro and in vivo conditions. We found that HM-10/10 peptide prevents oxidative stress induced in human fetal RPE (hfRPE) cells. Further, we show that HM-10/10 peptide protects RPE and retinal for sodium iodate (NaIO_3_)-induced geographic atrophy as well as in a retinitis pigmentosa model of photoreceptor degeneration by *N*-methyl-*N*-nitrosourea (MNU). Our studies indicate that HM-10/10 could be pursued as a novel therapeutic agent for treating retinal disease.

## 2. Results

### 2.1. HM-10/10 Peptide Significantly Inhibited Cell Death from tBH Treatment of hfRPE Cells

We examined the effect of HM-10/10 peptide on apoptosis in cell culture models. We first conducted dose response studies for HM-10/10 in hfRPE cells (Appendix A) and based on the results we used HM-10/10 at 10 μg/mL concentration in all the hfRPE cell culture experiments. hfRPE cells were cultured overnight in 0.5% serum containing DMEM medium and were either treated with HM-10/10 alone at 10 μg/mL or with 200 μM *tert*-Butyl hydroperoxide (*t*BH) [8,9] alone or with a combination of *t*BH and HM-10/10 for 24 h. Apoptosis was analyzed by TUNEL staining and as shown in Figure 1, *t*BH caused a significant increase in TUNEL positive cells in hfRPE and HM-10/10 significantly inhibited cell death of hfRPE cells, and also significantly inhibited apoptotic cell death induced by *t*BH (*p* = 0.002).

### 2.2. HM-10/10 Peptide Significantly Reduced Apoptotic Cell Death from NaIO_3_ Treatment of hfRPE Cells 

We next examined the effect of HM-10/10 peptide on apoptosis in NaIO_3_-induced hfRPE cell cultures. We performed a time course of NaIO_3_-induced cell death (Appendix A) and chose 24 h time point for subsequent experiments. As shown in Figure 2, treatment of hfRPE cells with either 200 µg/mL or 500 µg/mL of NaIO_3_ resulted in increased TUNEL-positive apoptotic cells; the induction of cell death was much higher with 500 ug/mL NaIO_3_ as compared to 200 μg/mL dose (Figure 2). Co-treatment with HM-10/10 peptide (10 µg/mL) at either concentration of NaIO_3_ significantly reduced apoptotic cell death (*p* = 0.003 or *p* < 0.0001).

### 2.3. HM-10/10 Peptide Protected hfRPE Cells from Oxidative Stress by Inhibiting Activation of Caspase-3/7

To further examine the apoptotic mechanism, we analyzed the activation of Caspase-3/7 in hfRPE cells using both the IncuCyte live imaging assay and Western Blot analysis (Figure 3). We used the *IncuCyte^®^* Caspase-3/7 Green *Apoptosis Assay* to determine the effect of HM-10/10 peptide on apoptosis in hfRPE cells. The green object counts of dead cells were increased when the cells were treated with either *t*BH or NaIO_3_ alone, but were significantly reduced when co-treated with HM-10/10 (*p* < 0.05) (Figure 3A). Western blot analysis showed cleaved Caspase-3 activation at 24 h was elevated in hfRPE cells treated with 200 μM *t*BH or NaIO_3_ alone, whereas co-treatment with HM-10/10 inhibited activation of Caspase-3 (Figure 3B).

### 2.4. Optimizing NaIO_3_ Doses In Vivo

We next sought to determine the protective effects of HM-10/10 peptide on retinal damage in vivo. HM-10/10 and other HDL mimetic peptides have been tested orally in C57BL6/J mice (unpublished data). NaIO_3_ treatment has been successfully used by others and us to study RPE degeneration in geographic atrophy in animal models [10,11]. Since previous dose response studies in NaIO_3_-induced RPE atrophy was performed in 129S6/SvEv and TAC mice [10,11], we first conducted a NaIO_3_ dose-response experiment to test the efficacy of HM-10/10 in C57BL6/J male mice. These experiments were conducted using doses of NaIO_3_ at 15, 20, and 35 mg/kg body weight (BW) administrated via tail vein injections to C57BL6/J male mice. Mice were fed a standard chow diet and after one week the extent of retinal damage was determined from optical coherence tomography (OCT) images by measuring overall retinal thickness (see Appendix A for comprehensive image analysis). Retinal thickness decreased significantly with increasing dose of NaIO_3_. Based on this finding, we chose a NaIO_3_ dose of 20 mg/kg BW in our subsequent experiments.

### 2.5. In Vivo Assessment of Retinal Structure Using Optical Coherence Tomography 

C57BL6/J male mice were given an intravenous injection of NaIO_3_ at 20 mg/kg BW or the vehicle PBS via tail vein. Mice were fed either a standard chow diet or a chow diet containing HM-10/10 at 100 mg/kg BW. OCT images were taken every week following the start of the treatment up to three weeks. Representative OCT and en face infrared fundus images for all the treatments are shown in Figure 4A. The mice receiving NaIO_3_ alone demonstrated a disruption in retinal structure, particularly in the outer retinal layers and RPE, with overall thinning of the retina. H&E staining established that the outer retinal layers are disorganized with discrete areas of debris in the RPE and outer retina consistent with outer retinal cell death. As a measure of damage to the retina, overall thickness measures were made from the OCTs using on-screen calipers positioned at discrete locations on either side of the optic nerve head. These thickness measures, shown in Figure 4B, demonstrate a thinning of the retina even after one-week exposure to NaIO_3_. The mice given NaIO_3_ plus the peptide in their chow also demonstrated an overall thinning of the retina. However, the addition of the peptide to the chow mitigated the toxic effect of NaIO_3_ and overall thinning was not as severe as in the group receiving NaIO_3_ alone (Figure 4B). Histological analysis of the retina after three weeks of treatment demonstrated findings consistent with in vivo OCT analysis. NaIO_3_ treatment caused a significant decrease in the number of nuclei in the outer nuclear layer (ONL) compared with the control mice. There was no significant difference in ganglion cell layer (GCL) or inner nuclear layer (INL) cell nuclei, but a significant difference in the ONL across all conditions. Importantly, HM-10/10 in the diet significantly improved the structure of the ONL compared to the group without the peptide, but did not restore ONL thickness to control values (Figure 4C). The representative histopathology images are shown in Figure 4C. Immunohistochemistry (IHC) staining of cleaved Caspase-3 of retinal tissue sections after three weeks of treatment showed lesser intensity of staining for Caspase-3 in the RPE layer (NaIO_3_ + HM-10/10) fed with HM-10/10 at 100 mg/kg BW compared with mice that received NaIO_3_ alone (Figure 4D).

### 2.6. HM-10/10 Caused Protection of Photoreceptor Degeneration and Improvement of Visual Function in MNU Mouse Model of Retinal Degeneration

The MNU + HM-10/10 treatment protocol consisted of both pre- and post-treatments. The retinal thickness of H&E stained sections were quantified to determine HM-10/10 induced protective effects on photoreceptors. The number of nuclei in ONL of MNU group and MNU + HM-10/10 group was reduced after MNU administration when compared with the control group. However, the mean number of nuclei in ONL from MNU + HM-10/10 group was significantly increased compared with MNU group (*p* < 0.0001, Figure 5A). The overall retina for MNU+HM-10/10 group was significantly thicker than MNU group (*p* = 0.0014, Figure 5A) as expected. Furthermore, the TUNEL assay was performed to quantify the apoptotic activity in photoreceptors. No apparent TUNEL-positivity was detected in the retinas of the normal control group, while many TUNEL-positive cells were found in the retinas of MNU group. Most of the TUNEL-positive cells were concentrated in the ONL, suggesting that MNU toxicity induced extensive photoreceptor apoptosis in mice retinas. TUNEL-positive cells were also detected in the retinas of the MNU+HM-10/10 group. However, the number of TUNEL-positive cells in MNU+HM-10/10 group was significantly less than that in MNU group (*p* < 0.00001; Figure 5B), suggesting that HM-10/10 treatment could alleviate the MNU induced photoreceptor apoptosis. 

Representative in vivo OCT images are shown in Figure 5C for the three treatment groups. Both the MNU and the MNU+HM-10/10 mice demonstrated structural disruption in the outer retina, particularly in the outer nuclear layer and its corresponding outer segments (Figure 5C upper panel). To quantify these changes, global thickness measures were obtained for the RPE, the outer retina (from RPE to inner nuclear layer), the nerve fiber layer, and total retinal thickness, as described above. Consistent with the NaIO_3_ study groups, the overall retina was thinner in the MNU treatment groups compared to the control with a mitigating effect of the HM10-10 peptide. Further analysis revealed that the composite measure responsible for the overall thinning (and rescue) effect was the outer retina. Significant differences across the three groups were observed only for the outer retina—the thicknesses of the inner retina and the RPE were not significantly different across the three groups (Figure 5C lower panel, *p* = 0.001). Thus, the in vivo OCT analysis is consistent with the TUNEL assay suggesting that MNU toxicity induced massive retinal photoreceptor apoptosis in mice which was mitigated by the peptide treatment.

Retinal function was evaluated with electroretinography (ERG). Figure 5D shows representative ERG responses to single intensity flashes under scotopic and photopic conditions. Figure 5E shows representative intensity-response functions for each treatment group under dark-adapted conditions. The MNU treatment groups clearly show the weakest responses with better responses for the MNU + peptide group. The I-R functions were fitted with the Naka-Ruston equation to estimate V_max_, the saturated retinal response. The experimental groups that received an injection of MNU alone or received MNU+HM-10/10 peptide treatment in their chow showed significantly lower amplitude responses under scotopic and photopic conditions than that for the control group (*p* = 0.0009, *p* = 0.005, respectively) (Figure 5D). However, the responses from those that received MNU plus HM-10/10 peptide treatment in their chow were significantly better than those that received MNU alone as demonstrated by the average V_max_ (± 1 se) across the three treatment conditions (*p* < 0.0001) (Figure 5D). Thus, the toxic effects of MNU were mitigated and the functional consequences were less severe following the peptide treatment. 

## 3. Discussion

HM-10/10 is a novel member of a class of HDL-mimetic peptides. HM-10/10 is constructed by utilizing specific sequences from apoE and apoJ proteins, both of which have been used previously in other HDL-mimetic iterations [12,13]. The chimeric peptide AEM-28 [14] has an 18 amino acid apoA-I mimetic peptide attached to C-terminal end of the apoE part of HM-10/10. AEM-28 has been shown to regulate dyslipidemia and inflammation in animal models [14]. The apoJ part of HM-10/10 is a G * peptide that has been shown to confer anti-oxidant and anti-inflammatory properties in animal models [15]. We designed HM-10/10 to confer both the lipid clearing of AEM-28 and the anti-inflammatory properties of G * peptide.

ApoA-I and HDL-mimetic peptides have been investigated in many animal models of disease. ApoA-I mimetic peptide 4F has been tested in two human studies [16,17,18] by oral administration and by intravenous (IV) or subcutaneous (SQ) administration [19]. The two oral studies [16,17] reported efficacy with doses of ≥3.3 mg/kg BW; lower doses were ineffective. At the time that the IV and SQ studies were designed, it was thought that these peptides acted in plasma, so the key success factor would be plasma peptide levels. Since these peptides are very expensive to produce, and the IV and SQ routes yield much higher plasma levels, these studies used lower doses. Consequently, in the IV and SQ studies [19], the maximum dose tested was 1.43 mg/kg BW; a dose that was ineffective in the two oral studies [16,17]. As expected, IV or SQ administration achieved dramatically higher plasma peptide levels, but surprisingly efficacy was not achieved [19]. These seemingly conflicting results led us to complete a series of studies that quite unexpectedly revealed that the peptides work in the intestine rather than in the plasma [20,21,22,23,24,25,26,27]. Therefore, we decided to use HM-10/10 peptide via an oral route.

Murine NaIO_3_ model is widely used as a model of RPE degeneration that occurs in geographic atrophy. It results in reproducible patch retinal degeneration and can be studied in a wide variety of wild type and genetic knock out strains. The sequence of events leading to retinal damage after NaIO_3_ will depend on the source of NaIO_3_, methods of administration, animals (type, strain, and age), and time points analyzed after injection [10,11]. High doses (50–100 mg/kg) lead to rapid complete ablation of the outer nuclear and RPE layer [28,29] while lower doses (15–35 mg/kg/kg) of NaIO_3_ are associated with decreased visual function as well as focal RPE loss and minimal outer retinal damage [11,30,31]. We and others have shown that the RPE damage results from increased ROS levels both in vitro and in vivo [10,31,32,33]. Several oxidative stress regulatory pathways such as Akt phosphorylation, PTEN, Nrf2, and mTOR have been shown to be involved [32,34,35]. The mechanism of cell death in RPE with NaIO3 is through apoptotic pathways although necroptosis has also been reported [36]. Investigations on the participation of specific pathways linked to the protective action of HM-10/10 in the retina will be of value and are being actively pursued in our laboratory.

To our knowledge, this is the first report on the role of a HDL-mimetic peptide in retinal protection. The beneficial effects of HM-10/10 peptide in protecting RPE/retina may arise from a combination of mechanisms for the neutralization of the oxy-radicals generated by the oxidant of NaIO_3_. Mitochondria are the prime source for the production of reactive oxygen species [37]. An effect of HM-10/10 may likely involve improvement of cellular, particularly mitochondrial function in stressed RPE/retina. For example, HM-10/10 peptide could upregulate GSH and enzymes of GSH biosynthesis as well as detoxify superoxide enzymes such as MnSOD. Our previous work has shown the expression and localization of GSH metabolizing enzymes and redoxin family of proteins in RPE cells and their regulation by oxidative stress [38,39]. Further, the mechanism of protection from caspase activation and ensuing apoptotic cell death by HM-10/10 peptide may be mediated by mitochondrial GSH (mGSH). mGSH plays a significant role in cellular defense against pro-oxidants, the depletion of mGSH affects cell viability, either by predisposing cells to apoptosis or by modulating mitochondrial membrane potential and subsequent activation of caspases through regulation of redox pathways [40]. We have reported that short chain (20–24-mer) bioactive peptides offer neuroprotection by selective upregulation of mitochondrial GSH [39,41,42]. Moreover, we have previously shown that D-4F, an apoA-I mimetic peptide, inhibits proliferation, and tumorigenicity of epithelial ovarian cancer cells by upregulating the antioxidant enzyme MnSOD [6].

The effect of oxidative stress in causing tissue inflammation is well established. The involvement of inflammation in the pathogenesis of AMD is well known and inflammasomal activation is implicated in AMD mediated by various risk factors [43,44]. NLRP3-mediated RPE death has been intensively debated due to the complicated involvement caspase proteins in AMD pathogenesis. Recently Mao et al. [45] found that mesenchymal stem cells protected NaIO_3_ triggered RPE death by deactivating NFkappaB mediated NLRP3 inflammasome and mitochondrial integrity. Our present data show that the activation of caspase 3/7 by NaIO_3_ in hfRPE is inhibited by HM-10/10 thereby preventing apoptotic cell death. Diverse AMD stressors may jointly influence and determine the ultimate cell fate via different pathways and thus it is conceivable that in addition to caspase 3, caspase 1 and NLRP3 inflammasome are activated with NaIO_3_ and HM-10/10 may inhibit these pathways.

Whether HM-10/10 peptide could mitigate cell death in vivo was assessed using OCT in the NaIO_3_ mouse model. Mice given an intravenous injection of NaIO_3_ demonstrated a significant thinning of the retina with heterogeneous disruption of the outer retina and RPE by OCT analysis, as previously reported [10,11]. However, those for which the HM-10/10 peptide was added to their chow showed significantly less disorganization of the outer retina, and less overall retinal thinning by OCT and immunohistochemistry, but did not completely rescue the normal phenotype (see Appendix A).

In addition to determining the beneficial effect of HM-10/10 peptide in dry AMD, we also investigated its effect in an experimental model of retinitis pigmentosa. MNU is known to induce concentration-dependent retinal degeneration mostly in the outer nuclear layer [46,47]. Retinitis pigmentosa is a blinding human disease that is characterized by the progressive loss of photoreceptor cells. Therefore, MNU-treated animals might serve as a model to study disease-associated features [47,48]. Both caspase dependent and caspase independent pathways for photoreceptor (PR) degeneration have been reported [47,48]. In addition, Endoplasmic Reticulum (ER) stress has been identified as another important cause of MNU-induced PR damage [49].

We found that MNU treatment gave evidence for selective loss of PR cells in the ONL as determined by apoptosis using TUNEL assays (Figure 5). Further, the consequences on retinal function were assessed by ERG. Treatment with MNU reduced retinal function to near non-detectable levels except at the brightest flash intensities, consistent with severe disruption of the outer- and middle-retina. When HM-10/10 peptide was added to their chow, retinal function was better than those mice that did not receive the peptide (Figure 5). Overall, in vivo assessments of retinal structure and function demonstrated improvements in HM-101/10 treated retinas, but did not restore the normal phenotype.

Finally, it is well known that the damage to mitochondria with oxidative, environmental, and genetic factors can lead to damage to mitochondrial DNA. Evidence implicates mitochondrial damage in the AMD disease process [37,50]. A potential link between mitochondrial dysfunction due to increased mitochondrial lesions and AMD has been reported [51,52]. Thus protecting DNA via therapeutics targeted to mitochondria in early in AMD could ameliorate or stop the progress to vision loss. Our results demonstrate that HM-10/10 peptide protects RPE and retina from oxidative damage in experimental models of geographic atrophy and retinitis pigmentosa suggests that it may be a promising therapeutic candidate in the prevention of AMD. 

## 4. Materials and Methods

### 4.1. Mice

Six-week-old C57BL6/J male mice were purchased from The Jackson Laboratory (The Jackson Laboratory, Bar Harbor, ME, USA). The Animal Research Committee at the University of California Los Angeles approved the mouse experimental protocols. All procedures used in this study were conducted in accordance with National Institutes of Health guidelines and the Association for Research in Vision and Ophthalmology Statement for the Use of Animals in Ophthalmic Vision Research. UCLA approved animal protocol 2004-161-33A was used. The title of the protocol is “The role of ApoAI and its peptides on inhibiting age-related pro-inflammatory diseases including cancer and retinal degeneration”. The approval period is valid till 19 May 2020.

### 4.2. HM-10/10 Peptide

HDL mimetic, HM-10/10 peptide (L-R-K-L-R-K-R-L-L-R-L-V-G-R-Q-L-E-E-F-L) was synthesized from all L-amino acids. The peptide had >98% purity, as determined by high-performance liquid chromatography (HPLC) and mass spectrometry (MS). The peptide was dissolved in H_2_O for cell culture and the administration in a chow diet for in vivo studies. The peptide was mixed into standard mouse chow (Ralston Purina) using techniques as described previously [19]. Each evening, the frozen diet was provided. The mice consumed all of the diet by next morning.

### 4.3. hfRPE Cell Culture Experiments

Early passage [2,3,4] primary cultured human fetal retinal pigment epithelial cells were cultured in 10% FBS DMEM medium [53]. Media was changed to 0.5% FBS DMEM the day before the treatment. Cells were treated with 200 uM *tert*-Butyl hydroperoxide (Sigma, St. Louis, MO, USA) or NaIO_3_ (200 μg/mL or 500 μg/mL) (Sigma, St. Louis, MO, USA) for 12 h or 24 h with or without the co-treatment of HM-10/10 at 10 μg/mL. hfRPE cells were cultured in glass bottom microwell dishes (MatTek, Ashland, MA, USA) and fixed in 4% paraformaldehyde solution and rinsed with PBS. Apoptotic cell death was measured using TUNEL assay according to the manufacturer′s instructions (Roche, Indianapolis, IN, USA). The number of TUNEL-positive cells was counted from eight different areas of each treatment under a fluorescent microscope and the average number of apoptotic cells was recorded. By using the IncuCyte live-cell analysis system (Essen BioScience, Ann Arbor, MI, USA), hfRPE cells were cultured in 96-well plates and imaged every hour up to 24 h after treatment to measure cell health and viability in real time. The data was analyzed by using the IncuCyte S3 software (Essen BioScience, Ann Arbor, MI, USA). Experiments were performed in quadruplicate; data were expressed as the mean of the quadruplicate determinations (mean ± SD).

### 4.4. Western Blot Analysis

Total cell proteins were collected at the end of indicated treatments in cell lysis buffer containing 0.1 M NaCl, 5 mM EDTA, 50 mM sodium orthovanadate, 1% Triton X-100, and protease inhibitor tablet in 50 mM Tris buffer (pH 7.5). 50 μg of total proteins were separated by SDS-PAGE and transferred onto nitrocellulose membrane, and followed by incubation with primary antibody at 4 °C in 5% skim milk and 0.1% Tween-20. Anti-cleaved Caspase-3 antibody (Cell Signaling, Danvers, MA, USA) was used at the dilution of 1:1000, and anti-GAPDH polyclonal secondary antibody was used at 1:5000.

### 4.5. Immunohistochemistry (IHC) Staining

Frozen retinal tissue sections were fixed with cold acetone at −20 °C for 10 min; and then blocked with 10% normal serum and 4% BSA prepared in PBS for three hours; then incubated with 1:500 rabbit polyclonal anti-cleaved Caspase-3 antibody (Cell Signaling, Danvers, MA, USA) overnight at 4 °C. Following washes, the sections were incubated with biotinylated secondary antibody (Jackson ImmunoResearch Laboratories, West Grove, PA, USA) for one hour at room temperature, followed by incubation with Vectastain ABC Elite reagents (Vector Laboratories. Burlingame, CA, USA).

### 4.6. TUNEL Assay

The TUNEL (terminal deoxyuridine triphosphate nick-end labeling) assay was performed on frozen retinal sections to analyze the apoptotic status of the retinas. The In Situ Cell Death Detection Kit (Roche Diagnostics, Indianapolis, IN, USA) was used according to the manufacturer′s protocol. The TUNEL sections were counterstained with DAPI, mounted on slides, and then visualized with fluorescence microscopy (Leica Camera Inc., Allendale, NJ, USA). The apoptotic cells/mm^2^ on ONL was calculated ([number of TUNEL-positive nuclei]/[mm^2^ Area of ONL]).

### 4.7. Spectral Domain Optical Coherence Tomography (SD-OCT) and Fundus Images

Ultra-high resolution spectral domain optical coherence tomography (SD-OCT) imaging was performed on both eyes for all groups, at one, two and three weeks post treatment or at the end of experiments, using a Bioptigen SD-OCT system (Research Triangle Park, Durham, NC, USA). A series of 100 b-scans were collected, stacked and aligned spatially to form a registered three-dimensional rendering of retinal volume. A high resolution horizontal *b*-scan centered on the optic nerve head was captured by averaging and spatially aligning 20 individual *b*-scans along the same horizontal axis. On screen calipers were placed at three equidistant locations on either side of the optic nerve head spanning the entire OCT image. At each location the total retinal thickness, defined as the distance (in μm) from the retinal pigment epithelium to the inner limiting membrane, was measured. The six measurements were averaged to provide a composite retinal thickness measure for each eye. Fundus images were obtained using the Micron II retinal imaging microscope (Phoenix Research Laboratories, Inc., CA, USA). For both OCT and fundus imaging, mice were anesthetized with an intraperitoneal (IP) injection of normal saline solution containing ketamine (15 mg/g body weight) and xylazine (7 mg/g body weight) and pupil dilation was accomplished by adding a drop of 1% atropine sulfate. The mouse was placed on a movable platform so that the eye could be aligned with the axis of the imaging device.

### 4.8. Electroretinography

Following overnight dark-adaptation, mice were anesthetized with an intraperitoneal injection of saline containing ketamine (15 mg/kg body weight) and xylazine (7 mg/kg body weight). ERGs were recorded from the corneal surface after pupil dilation (1% atropine sulfate) using a gold loop corneal electrode together with a mouth reference and tail ground electrode. A drop of methylcellulose (2.5%) on the corneal surface was used to ensure electrical contact and to maintain corneal integrity. Body temperature was maintained at 38 °C with a heated water pad. All stimuli were presented in a large integrating sphere coated with highly reflective white matte paint (Eastman Kodak Corporation, Rochester, NY, USA). Responses were amplified (Grass CP511 AC amplifier, × 10,000) and digitized using an I/O board (National Instruments, Austin, TX, USA) in a personal computer. Signal processing was performed with custom software LabWindows/CVI (National Instruments, Austin, TX, USA). For each stimulus condition, responses were computer-averaged with up to 50 records averaged for the weakest signals. A signal rejection window was adjusted on-line to eliminate artifacts. All stimuli were presented at 1 Hz except for the brightest flashes where the presentation rate was slowed to 0.2 Hz. The intensity-response functions were analyzed to extract V_max_, the maximum saturated b-wave amplitude.

### 4.9. Animal Models 

Two mouse models were used in this study: NaIO_3_ model of RPE atrophy and MNU model of photoreceptor degeneration.

#### 4.9.1. NaIO_3_ Model

At the beginning of in vivo studies, we performed a NaIO_3_ dose-response experiment with the C57BL/6J mice to identify the optimal doses to test the efficacy of our peptide. C57BL6/J male mice were treated with NaIO_3_ doses ranging from 15 mg/kg to 35 mg/kg BW administrated via tail vein injections. Mice were fed a standard chow diet and after one week the extent of retinal damage was determined from OCT images by measuring overall retinal thickness. 

Based on the dose-response data, we chose a NaIO_3_ concentration of 20 mg/kg for our subsequent experiments. Eighteen six-week-old C57BL6/J male mice were divided into three treatment groups consisting of six mice per group. The groups were: (1) control group (Control): Mice with tail vein injection of PBS (vehicle), fed with standard chow diet; (2) NaIO_3_-treated mice (NaIO_3_): Mice with tail vein injection of NaIO_3_ at 20 mg/kg BW [10], fed with standard chow diet; (3) NaIO_3_-treated mice with peptide (NaIO_3_ + HM-10/10): Mice received NaIO_3_ at 20 mg/kg via tail vein, fed with HM-10/10 peptide at 100 mg/kg body weight added to chow. Diet treatments were started on the same day the mice received the tail vein injections of PBS or NaIO_3_. At the end of each week, SD-OCT and fundus images were taken and analyzed. After three weeks of treatment, eyes were enucleated, and frozen sections of retinal tissue were used for histology. Retinal sections were scanned, the nuclei were counted from each layer and retinal thickness was quantified by Aperial Scan Scope (Leica Biosystems, Buffalo Grove, IL) using Aperio software as described previously [54]. 

#### 4.9.2. MNU Model

Retinal degeneration in MNU administered mice occurred within seven days with a dose of 60 mg/kg [55,56] which has been used in several therapeutic trials [11,12].

MNU solution at 5 mg/mL was prepared by dissolving reagent in physiological saline containing 0.05% acetic acid. The mice received a single intraperitoneal injection of MNU (Chem Service Inc., West Chester, PA, USA) at 60 mg/kg BW. Twelve mice were randomly assigned to three groups: (1) control group (Control): Mice received intraperitoneal (ip) injection of vehicle (physiological saline containing 0.05% acetic acid) and a standard chow; (2) MNU group (MNU): Mice received ip injection of MNU at 60 mg/kg BW and a standard chow; (3) MNU + HM-10/10 group (MNU + HM-10/10): Mice received ip injection of MNU at 60 mg/kg BW and fed with a diet containing HM-10/10 at 100 mg/kg BW added to chow. The mice received the regular chow or a chow containing HM-10/10 three days before MNU administration. On day five after MNU, SD-OCT images were taken and ERG function tests were analyzed. Eyes were enucleated, and frozen sections of retinal tissue were used for histology and TUNEL assay. Retinal sections were scanned, the numbers of nuclei were counted from each layer and retinal thickness was quantified. 

### 4.10. Statistical Analyses

The data are shown as means ± SD for each group. Statistical analyses were performed by the Student′s t–test, one-way or two-way ANOVA by Graph-Pad Prism 3.01. Difference between groups was established with Bonferroni′s post-hoc test. A *p*-value of less than 0.05 was considered statistically significant.

## Figures and Tables

**Figure 1 ijms-20-04807-f001:**
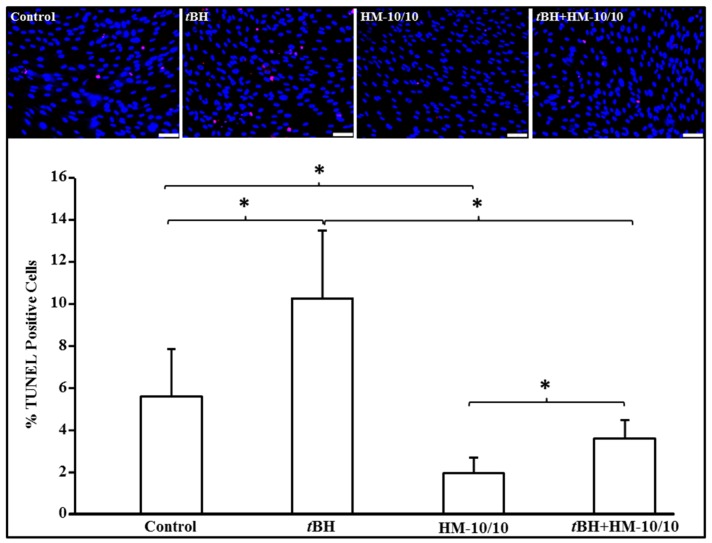
HM-10/10 peptide inhibits peroxide-induced apoptosis in human fetal retinal pigment epithelium (hfRPE) cells. hfRPE cells were incubated with HM-10/10 at 10 μg/mL alone or 200 μM *tert*-Butyl hydroperoxide (*t*BH) alone or the treated with a combination of *t*BH and HM-10/10 for 24 h as described under Materials and Methods. Apoptosis was analyzed and quantified by TUNEL staining (lower panel). Representative images are shown in the upper panel. White scale bar = 100 μm. Blue: DAPI nuclear staining; Red: TUNEL positive apoptotic nuclei staining. Asterisk indicates *p* < 0.008 with the Beonferroni correcition.

**Figure 2 ijms-20-04807-f002:**
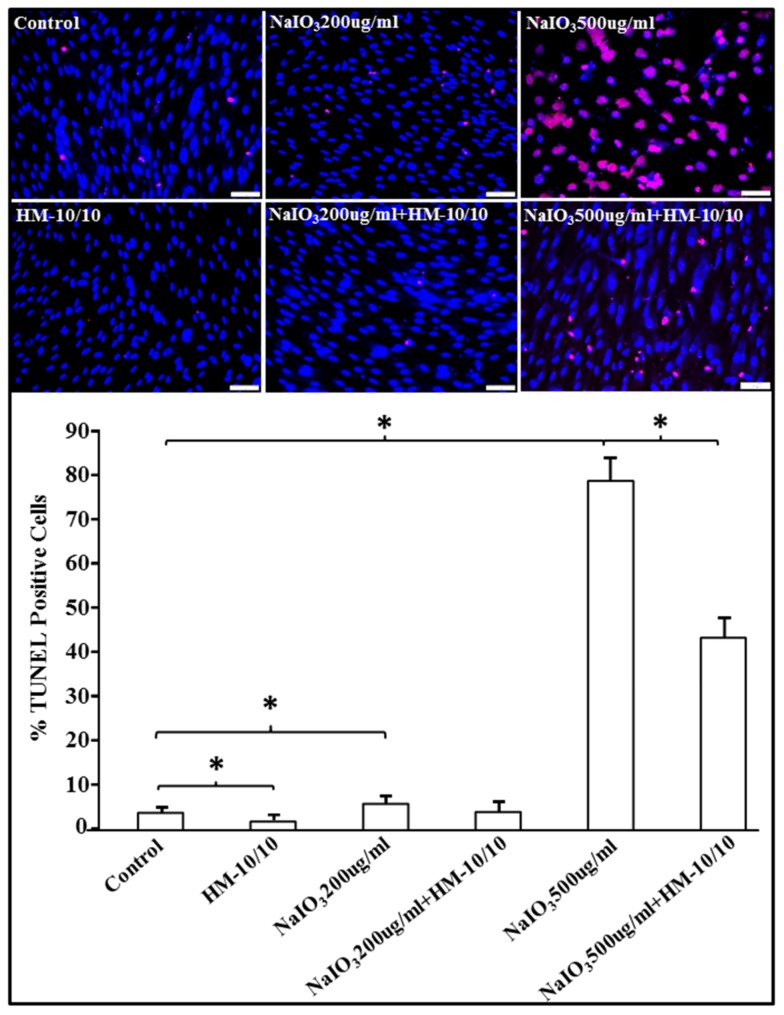
HM-10/10 peptide mitigates sodium iodate (NaIO_3_)-induced increase in apoptosis of hfRPE cells. hfRPE cells were incubated with HM-10/10 peptide at 10 μg/mL, or NaIO_3_ at either 200 μg/mL or 500 ug/mL, or treated with NaIO_3_ and HM-10/10 together, for 24 h as described under Materials and Methods. Apoptosis was analyzed and quantified by TUNEL staining (lower panel). Representative images showing TUNEL-positive apoptotic nuclei (*red*) and DAPI (*blue*) in all groups (upper panel). White scale bar = 100 μm. Asterisk indicates *p* < 0.0036 with the Bonferroni correction.

**Figure 3 ijms-20-04807-f003:**
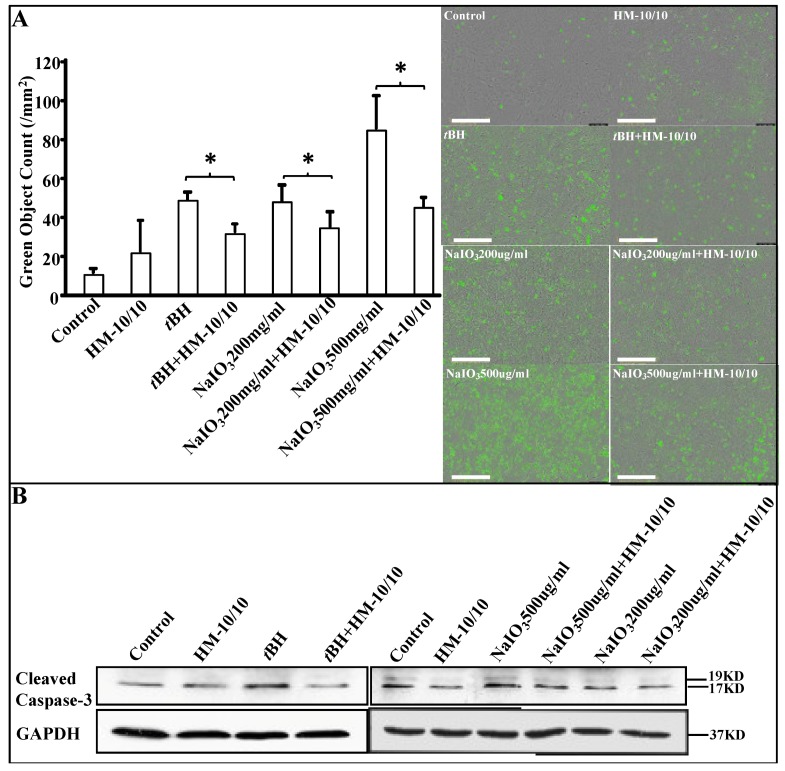
HM-10/10 inhibits the activation of Caspase-3/7 in hfRPE cells. hfRPE cells were incubated with HM-10/10 peptide at 10 μg/mL, *t*BH at 200 μM, or NaIO_3_ at either 200 μg/mL or 500 μg/mL alone, or treated with *t*BH and HM-10/10 together, or NaIO_3_ and HM-10/10 together, for 24 h as described under Materials and Methods. (**A**) The IncuCyte^®^ Caspase-3/7 Green Apoptosis Assay was used to determine the inhibitory effect of HM-10/10 peptide on cell proliferation of hfRPE cells. Representative images of the green object counts of dead cells are shown in the right panel. Automated real-time experiment by IncyCyte Zoom measured as green object count for all cells stained green with SYTOX Green. Magnification bar equals 400 μm. (**B**) The activation of Caspase-3 in hfRPE cells was determined by Western blot analysis. Asterisk indicates *p* < 0.05 with the Bonferroni correction.

**Figure 4 ijms-20-04807-f004:**
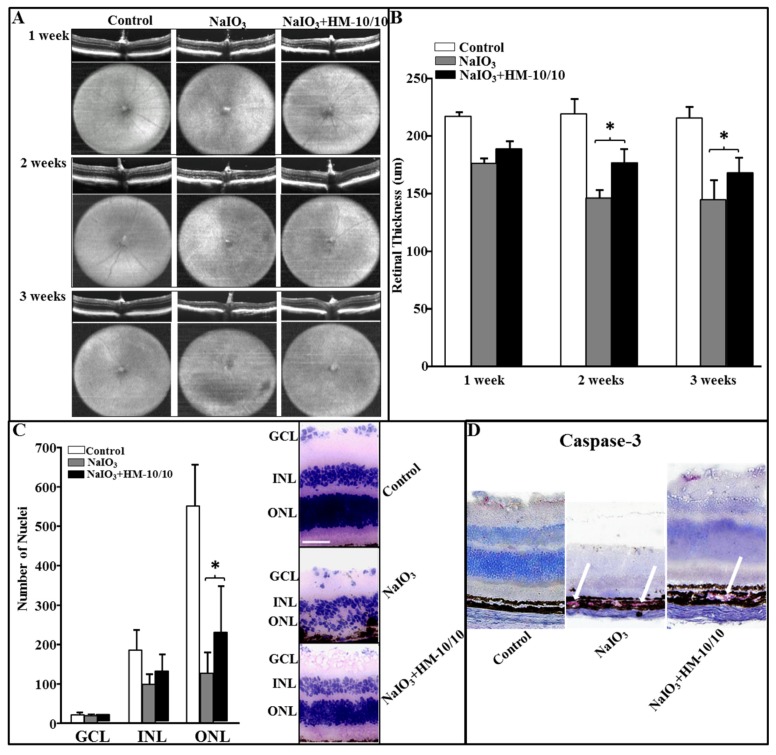
HM-10/10 treatment prevents degenerative changes in retina induced by intravenous NaIO_3._ Eye in vivo optical coherence tomography (OCT) images and the fundus photographs were taken after one, two, and three weeks of NaIO_3_ treatment as described under Materials and Methods. The mice were sacrificed at the end of three weeks. Retinal sections were scanned. (**A**) Degenerative changes shown by representative OCT and fundus images from Control, NaIO_3_, and NaIO_3_ + HM-10/10 groups of mice. Horizontal and vertical extents of fundus images are 1.4 × 1.4 mm, respectively. (**B**) Retinal thickness measurements from all three weeks of the study represented as bar graphs. (**C**) The numbers of nuclei were counted from each layer as described under Materials and Methods. The representative histopathology images showing decreased number of nuclei with NaIO_3_ treatment as compared to control and partial recovery in the number of nuclei with HM-10/10 treatment to that of control. Bar equals 50 μm. GCL: Ganglion cell layer; ONL: Outer nuclear layer; INL: Inner nuclear layer. (**D**) Immunohistochemistry staining of cleaved Caspase-3 (red) of retinal tissue sections. Bonferroni method was used to adjust the *p* < 0.0016 for multiple pairwise comparisons.

**Figure 5 ijms-20-04807-f005:**
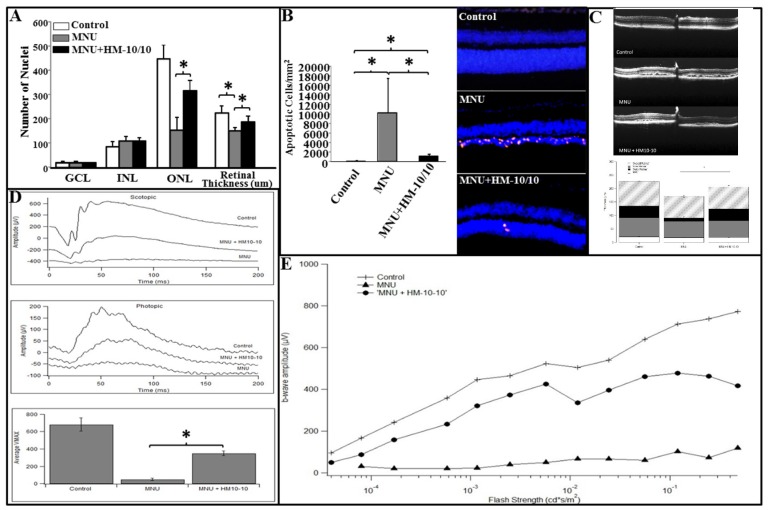
HM-10/10 protective effects on photoreceptors and on visual function in MNU mouse model. C57BL6/J male mice were injected with MNU via intraperitoneal injection and received pre- or post-treatment of HM-10/10 in chow as described under Materials and Methods. After the treatment, retinal damage was analyzed and electroretinography (ERG) retinal function tests were assessed. (**A**) The numbers of nuclei were counted from each layer and retinal thickness was measured as described under Materials and Methods. (**B**) TUNEL assay was performed to quantify the apoptotic activity in retinas. (**C**) Representative OCT cross-sectional image overlapped at the optic nerve head of the retina depicting the architectural trauma of the outer retina (upper panel) and overlapped bar graphs comparing the overall retinal thickness measurements to the composite three cell layers (lower panel). Horizontal extent of OCT cross-sections is 1.4 mm. (**D**) Representative waveforms of the brightest stimulus from each group under scotopic (0.2 cd·s/m^2^) and photopic (6.0 cd·s/m^2^) conditions (upper two panels). Averages and SEM of the saturated retinal response amplitudes by group (lower panel). (**E**) Representative I-R functions for each group according to flash strength. Asterisk indicates *p* < 0.0016 with the Bonferroni correction.

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
