# Peer review of "A Novel HDL-Mimetic Peptide HM-10/10 Protects RPE and Photoreceptors in Murine Models of Retinal Degeneration"

_ijms, 2019, doi:10.3390/ijms20194807_

Round 1

Reviewer 1 Report

The authors replied well to all my comments. I recommend publication of the study in its current form.

Reviewer 2 Report

 All the comments have been addressed.

This manuscript is a resubmission of an earlier submission. The following is a list of the peer review reports and author responses from that submission.

Round 1

Reviewer 1 Report

The study is devoted to the cell culture (primary cultured human fetal retinal pigment epithelial cells, hfRPE) and animal model (mice) characterization of the potential of HM-10/10 peptide to treat retinal diseases. HM-10/10 is chimeric HDL- mimetic peptide with anti-oxidant properties. The design of the peptide is thoroughly motivated based on the role of HDL peptides for lipid homeostasis and their anti-oxidant, anti-inflammatory and immunomodulatory properties. HM-10/10 peptide increased the hfRPE cells resistance to exposure of tert-Butyl hydroperoxide and NaIO3 tand ameliorated photoreceptor cell death and improved retinal function in mouse model with MNU-induced photoreceptor degeneration. The cell biology and the animal physiology techniques are thoroughly described and the results are properly presented and analyzed.

My recommendation considers better description of the statistical analysis. Please specify the software used. Also point 4.10. states that unpaired t-test is utilized. Most of the figures feature three and more groups of data. In case of multiple comparisons the inflation of false positive rates should be prevented. Thus Analysis of Variance with post-hoc tests is normally preferred. The implementation of t-tests is also possible but with Bonferroni correction (other corrections are also possible). Please describe the methodology in more detail.

A minor technical comment: at line 432 remove the red underline of NaIO3 in the screen captured Supplementary figure.

Reviewer 2 Report

This manuscript demonstrates the beneficial effect of an HDL mimetic peptide, HM-10/10, in in-vitro hfRPE cells and in-vivo experimental murine models of retinal degeneration. The work shows that HM-10/10 delivered orally protects the degenerating photoreceptors and the outer retinal layers and thereby improves visual function, highlighting HM-10/10’s possible role in treating retinal diseases.   

My comments and suggestions are as below.

Line 65, 71, 72, Figure 1: A dose response curve (with respect to time and concentration) of HM-10/10 on hfRPE cells is needed. Line 65: reference needed for tBH treatment Line 124-125: Why was the treatment only for 3 weeks? Was extended treatment given to rule out any adverse effects? Figure 4C: Please include representative images of retinal sections from control, NaIO3 and NaIO3 + HM-10/10 groups to show the number of nuclei. HM-10/10 was administered via an oral route in rat chow @ 100mg/kg. What was the available intravitreal concentration? What is the half life for HM10/10 in the body? The authors should explain in brief how orally administered peptide drugs reach the internal regions of the eye ie the RPE and the retina (systemic to ocular site of action)?     Studies in an aging or inherited murine model of retinal degeneration will strengthen the manuscript and link it to AMD.